

# Boosting determinant quantum Monte Carlo with submatrix updates: Unveiling the phase diagram of the 3D Hubbard model

**Fanjie Sun[1] and Xiao Yan Xu[1,2⋆]**

**1** Key Laboratory of Artificial Structures and Quantum Control (Ministry of Education),
School of Physics and Astronomy, Shanghai Jiao Tong University,
Shanghai 200240, China
**2** Hefei National Laboratory, Hefei 230088, China

⋆ xiaoyanxu@sjtu.edu.cn

## Abstract

Determinant Quantum Monte Carlo (DQMC) provides numerically exact solutions for strongly correlated fermionic systems but faces significant computational challenges with increasing system size. While submatrix updates were originally developed for Hirsch-Fye QMC with onsite interactions at finite temperatures [1], their comprehensive application in DQMC has remained unexplored despite noted algorithmic similarities. We present the first comprehensive application of submatrix updates in DQMC, significantly extending beyond the original scope by enabling simulations with extended interactions and at zero temperature. Building upon conventional fast updates and delay updates, our generalized implementation achieves an order-of-magnitude improvement in computational efficiency, enabling simulations of the half-filled Hubbard model on lattices up to 8,000 sites - a scale previously challenging with standard DQMC implementations. This enhanced computational capability allows us to accurately determine the finite-temperature phase diagram of the 3D Hubbard model at half-filling. Our findings not only shed light on the phase transitions within these complex systems but also pave the way for more effective simulations of strongly correlated electrons, potentially guiding experimental efforts in cold atom simulations of the 3D Hubbard model.

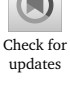

# 1  Introduction

In recent years, theoretical research on strongly correlated fermionic systems has become a focus in the field of condensed matter physics. With the proposal of various approximate models for such systems and the rapid development of computer technology, numerical computational methods have gathered increasing attention in this domain. Among various numerical methods, the exact diagonalization (ED) method [2] stands out as a universal approach capable of accurately determining system properties. However, its computational complexity exhibits an exponential increase with the size of the system, posing a significant challenge for precise treatment of large systems. The density matrix renormalization group (DMRG) method [3,4] demonstrates accurate solutions for one-dimensional systems, but it requires exponentially large bond dimensions due to the area law or even faster increasing of entanglement for the higher dimensional systems. The tensor network (TN) method [5], which naturally reflects the entanglement structure has been achieving promising progress, but its computational complexity is still a very high power of bond dimensions. The dynamical mean field theory (DMFT) [6] excels in combining with local-density approximation (LDA) for material calculations, but inherently loses long-range spatial fluctuations, and may encounter challenges in converging to physically meaningful solutions under strong correlation conditions [7,8], thereby compromising computational accuracy. The quantum Monte Carlo (QMC) method stands out as a numerically exact method, however, it usually suffers from the sign problem.

For models free from the sign problem, the QMC method is expected to accurately simulate very large system sizes. This is indeed the case for quantum spin systems, for example, Stochastic Series Expansion (SSE)-QMC method [9] can typically handle hundreds of thousands of sites. However, for fermionic systems, the simulation becomes much heavier. For example, determinant QMC (DQMC) [10–18], which is usually also called Blankenbecler-Scalapino-Sugar (BSS) method [10] or auxiliary field QMC (AFQMC), has found widespread application in solving strongly correlated fermionic systems, yielding numerous meaningful scientific results [19–46]. Despite its significant success, the DQMC algorithm still faces substantial limitations. Due to the tedious processing of a large number of Green's function matrices, including updates and matrix multiplications during the simulation, the DQMC algorithm

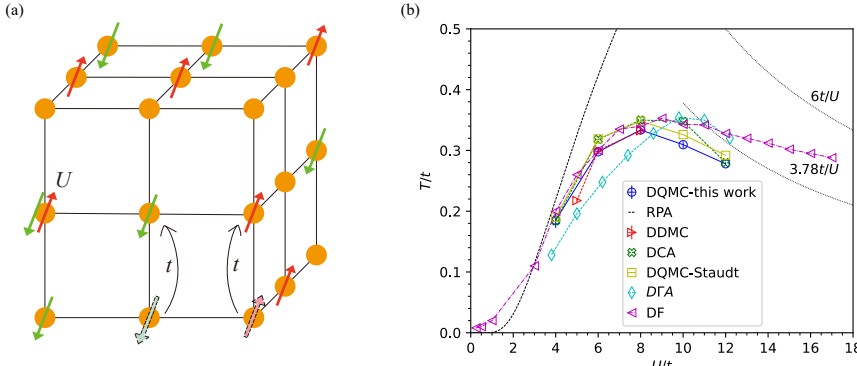

Figure 1: (a) The Hubbard model on a three-dimensional cubic lattice. The parameter $t$ describes the hopping of electrons between nearest neighbor (NN) sites $i$ and $j$, while $U$ represents the amplitude of the onsite Hubbard repulsive interaction. (b) The phase diagram of the Néel transition for the three-dimensional half-filled Hubbard model on a cubic lattice, as obtained by DQMC in this work (blue circles). The comparison curves are obtained by the random phase approximation (RPA) from Refs. [48] (dashed black line), determinantal diagrammatic Monte Carlo (DDMC; red right triangles) [49], dynamical cluster approximation (DCA; green X-shaped) [50], DQMC-Staudt's work (DQMC; yellow squares) [51], dynamical vertex approximation (DΓA; light blue rhombus) [52], the dual-fermion multiscale approach (DF; purple left triangles) [53]. The $T_c/t = 3.78t/U$ line is the Heisenberg limit [54, 55]. The $T_c/t = 6t/U$ line is from a mean field estimation [56].

exhibits high computational complexity $\mathcal{O}(\beta N^3)$, where $N$ is the total number of lattice sites, $\beta$ is the inverse temperature. This complexity and its huge prefactor together make it challenging to simulate very large system size, thus usually one cannot faithfully extrapolate the limited finite size results to the thermodynamic limit and obtain accurate phase transition properties. For instance, in the simulation of interacting Dirac fermion systems with Gross-Neveu transition, the critical exponents do not converge among different simulations on the same universality class [27,32,47]. Among those simulations, the typical system size is around 1000 sites, and only a few examples approach about 2500 sites [32]. Another example is the three-dimensional half-filled Hubbard model, many research groups have analyzed its phase diagram using various methods [48–56], current applications of the DQMC algorithm are limited to systems with only around 1000 lattice sites [51]. This limitation poses challenges in accurately determining the temperature for the Néel antiferromagnetic phase transition.

Therefore, increasing the efficiency of simulating fermionic systems is an urgent task. One promising direction relies on the developing Hamiltonian lattice field theories [57], with a large finite temporal lattice spacing, a simulation of 10,000 sites is achieved, and a simulation of 4,096 sites in the small temporal lattice spacing limit is obtained. Another attempt comes from using hybrid Monte Carlo (HMC) to study strongly correlated fermionic systems [58], which is expected to have lower computational complexity, but it turns out there are ergodicity issues due to zeros of the determinant in models such as Hubbard model. Algorithms designed to avoid these ergodicity issues may introduce high computational complexity [58] or explicitly symmetry breaking [47, 59–61]. For the long range coulomb coupling problem, when the fermion determinant is not too ill-conditioned, HMC can simulate amazing large system size with more than 20,000 lattice sites [62]. There is also encouraging progress in developing fermionic numerical methods without determinants [63], but their computational efficiency for large systems is still waiting to be tested.

Directly improving the efficiency of DQMC has always been a challenging problem. Based on self-learning Monte Carlo method [64, 65], it is able to improve the simulation system size up to 10,000 sites at very high temperature for a spin-fermion model, but the speedup strongly depends on the accuracy of the effective model, which is used to guide the update. Low-rank approximation also shows significant speedup for the low-density case [66], however, its theoretical accuracy for large densities is not well established theoretically. The polynomial expansion approximation based on sparse matrix multiplication enables zero-temperature DQMC calculations to simulate systems with up to 10,952 sites, but it may induce a truncation error [67]. Recently, we proposed a generalized delay update algorithm [68], originally proposed in Refs. [1,69] to study onsite Hubbard models at finite temperatures in Hirsh-Fye QMC and continuous-time QMC. We apply it in DQMC and have generalized it to study models with extended interactions at both finite and zero temperatures. Furthermore, there is still room left to improve the efficiency further by using the submatrix update [1], which has already been implemented in Hirsch-Fye QMC and continuous-time QMC to simulate the Hubbard model [1, 70]. In this article, we provide a first comprehensive application of the submatrix update method to the DQMC algorithm and used this method to calculate the phase diagram of the three-dimensional half-filled Hubbard model. Then, we generalize it to handle extended interactions as well as zero-temperature cases. With the help of the speedup gain from submatrix update, we are able to simulate system sizes up to 8,000 sites without pushing hard, thus being able to accurately determine the finite temperature phase diagram of the 3D Hubbard model at half-filling, as shown in Fig. 1. We anticipate this more accurate phase diagram will help to guide the cold atom simulation of the 3D Hubbard model [71].

In the following, we briefly compare three different update schemes used in DQMC, the conventional fast update, the delay update, and the submatrix update. In the DQMC algorithm, Trotter decomposition is employed to partition the inverse temperature $\beta$ into numerous small slices ($\beta = L_\tau a_\tau$, where $a_\tau$ is a small imaginary time slice and $L_\tau$ is the total number of imaginary time slices), while the Hubbard-Stratonovich (HS) transformation is utilized to decouple the interaction term by introducing auxiliary fields. Typically, there are $\mathcal{O}(\beta N)$ auxiliary fields (where $N$ represents the system size). We assume each auxiliary field connects $k$ spatial sites, and consider only when $k$ does not scale with system size, namely, we only consider interactions with locality. After those steps, the problem has been transformed to be a problem with fermion bilinears coupled to auxiliary fields, such that the fermion degrees of freedom can be exactly traced out, resulting in a determinant of the inverse of the equal-time Green's function matrix as the Boltzmann weight for each configuration of auxiliary fields. If any Boltzmann weight is semi-positive, which means there is no sign problem, we can interpret it as probability, such that Monte Carlo can be used to perform importance sampling. If there is a sign problem, we usually need to define a sign free referenced system to perform the Monte Carlo simulation. In this case, the average sign usually shows an exponentially decay with system size, resulting in an exponentially scaling of complexity to achieve a controllable error bar. Of course, there are exceptions. It was first observed in a simulation that the average sign could algebraically decay with system size [72]. This phenomenon was later also found in several other cases [73]. Interested readers can refer to Refs. [72–74] for more details. Here, let's focus on cases free from the sign problem and note that all the update schemes discussed here also apply to cases with a sign problem.

During the simulation, we try to flip auxiliary fields one by one by performing local updates. We define *a sweep* as traversing auxiliary fields across all spatio-temporal sites in a sequence that starts from an initial site, progresses to the final site, and then reverses back to the initial site. In the following, when we talk about the computation complexity, by default we talk about the complexity per sweep. When flipping an auxiliary field at a specific spatio-temporal site, the proposed flip alters only a small region of the coefficient matrices for fermion bilinears.

Table 1: The computational complexity and types of computations required for various local updates. Here, 'update-ratio' refers to the calculations needed to obtain intermediate matrices/vectors for calculating the determinant ratio and to accumulate vectors used to finally update the Green's function, and 'update-G' refers to the calculations for updating the entire Green's function. 'Level 1' means Level 1 BLAS, and 'Level 3' means Level 3 BLAS. See Sec. 2 for more details.

|  | fast update | delay update | submatrix update |
|---|---|---|---|
| update-ratio | - | $\mathcal{O}(\beta n_d N^2)$ | $\mathcal{O}(\beta n_d^2 N)$ |
|  |  | Level 1 | Level 1 |
| update-G | $\mathcal{O}(\beta N^3)$ | $\mathcal{O}(\beta N^3)$ | $\mathcal{O}(\beta N^3 + \beta n_d N^2)$ |
|  | Level 1 | Level 3 | Level 3 |

Consequently, the new Green's function matrix differs from the old one by a low-rank (rank $k$) matrix. Therefore, the calculation of the ratio of the determinant can be simplified by using Sylvester's determinant theorem, resulting only $\mathcal{O}(k^3)$ complexity to calculate the determinant ratio. If the proposed flip is accepted, we update the full Green's function by using the fact that the new one and the old one only differ by a low rank (rank $k$) matrix. In the code, this can be done by Level 1 Basic Linear Algebra Subprograms (BLAS). This is so called *fast update*. The fast update scheme can be further improved by using delay update. The update of full Green's function can be delayed as the calculation of the determinant ratio only requires a small part of the Green's function, therefore we can store the intermediate vectors which will be used to calculate the determinant ratio for each local update, and finally be used to update the full Green's function after $n_d$ number of vectors are accumulated, where $n_d$ is determined by test in practice, and a suggested value is presented in the detailed discussion of the algorithm in the following. The final update of the full Green's function is done by Level 3 BLAS. This is the basic idea of *delay update*, and it makes better usage of cache by replacing Level 1 BLAS with Level 3 BLAS, which accelerates the calculation. However, using intermediate vectors to calculate determinant ratios will bring additional overhead, which has complexity $\mathcal{O}(\beta n_d N^2)$ per sweep. As this additional overhead is implemented with Level 1 BLAS, its time cost can surpass the time cost for the update of Green's function in practical calculation if one increase $n_d$, as shown in Fig. 3. The submatrix update can reduce this part to be $\mathcal{O}(\beta n_d^2 N)$, thus significantly reduce the computation with Level 1 BLAS. The computation complexity for all update scheme are summarized in Table 1.

The remaining part of the article is structured as follows: In Section 2, we introduce the basic formalism of the submatrix update method. In Section 3, we compare the efficiency of the submatrix update method with the delay update method proposed previously [68]. This comparison is conducted on both the Hubbard model and the spinless $t$-$V$ model on a two-dimensional square lattice. Additionally, we provide the phase diagram for the antiferromagnetic Néel order phase transition in the three-dimensional Hubbard half-filled model on a cubic lattice. Finally, we present a brief conclusion and discussion in Section 4.

## 2  Method

We first revisit the key aspects of the derivation presented in Ref. [68] within the framework of the fast update and the delay update. Then we introduce the submatrix updates for DQMC, outlining the notation and conventions. We focus on the finite temperature version of DQMC

(DQMC-finite-T) and address the zero-temperature case in Appendix D. Taking the Hubbard model [75–78] as an example, with Hamiltonian

$$H_{tU} = H_0 + H_U \,, \tag{1}$$

where $H_0 = -t \sum_{\langle i,j \rangle,\alpha}(c_{i,\alpha}^\dagger c_{j,\alpha} + \text{H.c.})$ and $H_U = \frac{U}{2} \sum_i (n_i - 1)^2$. Here, $c_{i,\alpha}^\dagger$ represents an electron creation operator on site $i$, where $\alpha = \uparrow/\downarrow$ denotes the spin polarization direction of electrons, and $n_i = \sum_\alpha c_{i\alpha}^\dagger c_{i\alpha}$ represents the electron occupation. The term $H_0$ describes the hopping of electrons between nearest neighbor (NN) sites $i$ and $j$, while $H_U$ accounts for the onsite Coulomb repulsion between two electrons.

The Hubbard model effectively captures the competitive relationship between the kinetic energy term associated with electron hopping and the potential energy term arising from Coulomb repulsion. In this study, we employ the following Hubbard-Stratonovich (HS) transformation to decouple the interaction term into a coupling between fermions and bosonic auxiliary fields:

$$e^{-a_\tau \frac{U}{2} \sum_i (n_i-1)^2} = \frac{1}{4} \sum_{s=\pm 1, \pm 2} \gamma(s) e^{i\alpha_U \sum_i \eta(s)(n_i-1)} + O(a_\tau^4) \,. \tag{2}$$

In the HS transformation, $s$ represents an auxiliary field, $\alpha_U = \sqrt{a_\tau U}$, and $\gamma(\pm 1) = 1 + \frac{\sqrt{6}}{3}$, $\gamma(\pm 2) = 1 - \frac{\sqrt{6}}{3}$, $\eta(\pm 1) = \pm\sqrt{2(3 - \sqrt{6})}$, $\eta(\pm 2) = \pm\sqrt{2(3 + \sqrt{6})}$. By tracing out the degrees of freedom of the fermion, the partition function can be expressed as follows:

$$Z = \sum_s w_b[s] w_f[s] \,. \tag{3}$$

For convenience, let's denote all auxiliary fields on all the spatio-temporal sites as $s$. The bosonic weight can be clearly defined as

$$w_b[s] \equiv \prod_{s \in s} \frac{1}{4} \gamma(s) e^{-i\alpha_U \eta(s)} \,. \tag{4}$$

And the fermion weight can be expressed in the form of a determinant

$$w_f[s] = \det[I + B_s(\beta, 0)] \,. \tag{5}$$

$B_s(\beta, 0)$ is the time evolution matrix from imaginary time $0$ to imaginary time $\beta$. A general time evolution matrix from imaginary time $\tau_1 = l_1 a_\tau$ to imaginary time $\tau_2 = l_2 a_\tau$ ($\tau_2 \geq \tau_1$) is defined as

$$B_s(\tau_2, \tau_1) = B_s^{l_2} B_s^{l_2-1} \cdots B_s^{l_1+1} \,, \tag{6}$$

where $B_s^l = e^{V(s)} e^{-a_\tau K}$ is the time evolution matrix for time slice $l$, where $K$ is the coefficient matrix of the fermion bilinear of the non-interacting part $H_0 = \sum_{j,k} c_j^\dagger K_{j,k} c_k = c^\dagger K c$, and $V(s)$ is the coefficient matrix of the fermion bilinear after HS transformation of the interacting part $i\alpha_U \sum_i \eta(s) n_i = \sum_{j,k} c_j^\dagger V(s)_{j,k} c_k = c^\dagger V(s) c$. Note we have omitted the spin index, as it can be absorbed into the site index. For the case like Hubbard model where both $K$ and $V(s)$ are block diagonal in spin space, one can split the determinant in Eq. (5) into a product of two determinants for each spin, such that the determinant ratio can be calculated separately for each spin. However, to maintain generality, we discuss a general case in what follows.

## 2.1 Fast update

Let's now consider the local update of the auxiliary field $s$. For a proposed update of auxiliary field at lattice site $i$ and imaginary time slice $l$, $s_{i,l} \rightarrow s'_{i,l}$, it leads to a change $e^{V(s)} \rightarrow e^{V(s')} = (I + \Delta(s, s'))e^{V(s)}$, so the change $\Delta(s, s')$ can be expressed as $\Delta(s, s') = e^{V(s')}e^{-V(s)} - I$, where $s'$ represents the proposed updated auxiliary fields. Then we calculate the acceptance ratio to determine whether to accept the proposed update. The determinant part ratio can be calculated in the following way:

$$\frac{w_f[s']}{w_f[s]} = \det\left[I + \Delta(s, s')(I - G_s(\tau, \tau))\right], \tag{7}$$

where $G_s(\tau, \tau)$ is the equal-time Green's function matrix, defined as $G_{s,ij}(\tau, \tau) \equiv \langle c_i(\tau)c_j^\dagger(\tau)\rangle_s$, and is related to the time evolution matrix [10–18],

$$G_s(\tau, \tau) = (I + B_s(\tau, 0)B_s(\beta, \tau))^{-1}. \tag{8}$$

Note that in order to keep the notations succinct, we will omit the auxiliary field dependence on $s$ and $s'$ in the time evolution matrix $B$, the Green's function $G$, the $\Delta$ matrix, and all other related matrices by default. We will only add back the dependence when necessary. Generally, one can utilize unitary transformations to convert $\Delta$ into a sparse matrix with $k$ non-zero diagonal elements. The unitary transformations can be absorbed into the time evolution matrix $B(\tau, 0)$ and $B(\beta, \tau)$. Let's assume that $\Delta$ is already transformed into its diagonal form and the positions of its $k$ non-zero diagonal elements are $x_1, x_2, \ldots, x_k$. Finally, we can apply Sylvester's determinant theorem to simplify the determinant ratio to:

$$\frac{w_f[s']}{w_f[s]} = \det[S], \tag{9}$$

where

$$S = I_{k \times k} + \mathcal{V}D \tag{10}$$

is only a $k$-dimensional matrix, as well as $I_{k \times k}$, $\mathcal{V}$ and $D$. $I_{k \times k}$ is an identity matrix, and $\mathcal{V}$ and $D$ are defined as

$$\mathcal{V} = \begin{bmatrix} -(G_{x_1 x_1} - 1) & -G_{x_1 x_2} & \cdots \\ -G_{x_2 x_1} & -(G_{x_2 x_2} - 1) & \cdots \\ \vdots & \vdots & \ddots \end{bmatrix}_{k \times k}, \tag{11}$$

$$D = \begin{bmatrix} \Delta_{x_1 x_1} & & \\ & \Delta_{x_2 x_2} & \\ & & \ddots \end{bmatrix}_{k \times k}. \tag{12}$$

Once the proposed update is accepted, in the fast update scheme, one utilizes the Woodbury matrix identity to update the entire Green's function matrix. The Green's function matrices before (denoted as $G$) and after (denoted as $G'$) the update only differ by a rank-$k$ matrix which can be represented as a product of three matrices: $\mathbb{U}\mathbb{S}\mathbb{V}$,

$$G' = G + \mathbb{U}\mathbb{S}\mathbb{V}, \tag{13}$$

where the matrix $\mathbb{U}$, $\mathbb{S}$ and $\mathbb{V}$ have dimensions $N \times k$, $k \times k$ and $k \times N$ respectively, and have the following form:

$$\mathbb{U} = [G_{:,x_1}|G_{:,x_2}|\cdots]_{N \times k}, \tag{14}$$

$$\mathbb{S} = DS^{-1}, \tag{15}$$

$$\mathbb{V} = \left([G_{x_1,:} - e_{x_1}|G_{x_2,:} - e_{x_2}|\cdots]^T\right)_{k \times N}, \tag{16}$$

where $e_{x_j}$ denotes a length-$N$ row vector with 'one' at position $x_j$ and zero at all other positions, $G_{:,x_j}$ denotes column $x_j$ of $G$, and $G_{x_j,:}$ denotes row $x_j$ of $G$. This concludes the basic formulation for the fast update method.

## 2.2  Delay update

For the delay update, we follow Ref. [68] and review the key steps. The delay update basically delays the entire Green's function matrix update. We store the intermediate $\mathbb{U}$, $\mathbb{S}$ and $\mathbb{V}$ matrices, and for any intermediate step $i$, the Green's function can be calculated with the following form

$$G^{(i)} = G^{(0)} + \sum_{m=1}^{i} \mathbb{U}^{(m)}\mathbb{S}^{(m)}\mathbb{V}^{(m)}. \tag{17}$$

However, in practice, we do not calculate the entire Green's function at every step $i$. Instead, we only calculate a small region of $G^{(i)}$ necessary for calculating the determinant ratio and the intermediate $\mathbb{U}^{(i)}$, $\mathbb{S}^{(i)}$, and $\mathbb{V}^{(i)}$ matrices. Finally, we calculate the entire Green's function matrix when $i = n_d$, with $n_d$ in principle depending on the computation platform. Based on our test over several different platforms [68], setting $kn_d = 2^\lambda$, where $\lambda = \max[6, [\log_2 \frac{N}{20}]]$, is a good choice.

## 2.3  Submatrix update

The submatrix update can further improve the efficiency [1, 69]. To facilitate the derivation of the submatrix update method, we rewrite the $i$-th update of the Green's function in the fast update method as equations (13) as follows:

$$G^{(i)} = G^{(i-1)} + \mathbb{U}^{(i)}\mathbb{S}^{(i)}\mathbb{V}^{(i)}, \tag{18}$$

$$\mathbb{S}^{(i)} = D^{(i)}(I_{k\times k} + \mathcal{V}^{(i)}D^{(i)})^{-1}. \tag{19}$$

For convenience, we use the inverse of the Green's function matrix to help the formulation. The inverse of the Green's function is defined as:

$$A^{(i)} \equiv (G^{(i)})^{-1}. \tag{20}$$

Using the Woodbury matrix identity, the Eq. (18) can be rewritten as

$$\begin{aligned}
A^{(i)} &= A^{(i-1)} - P_{N\times k}\left[x^{(i)}\right]D^{(i)}P_{k\times N}\left[x^{(i)}\right] + P_{N\times k}\left[x^{(i)}\right]D^{(i)}P_{k\times N}\left[x^{(i)}\right]A^{(i-1)} \\
&= \left[I + P_{N\times k}\left[x^{(i)}\right]D^{(i)}P_{k\times N}\left[x^{(i)}\right]\right]A^{(i-1)} - P_{N\times k}\left[x^{(i)}\right]D^{(i)}P_{k\times N}\left[x^{(i)}\right],
\end{aligned} \tag{21}$$

where $\left[x^{(i)}\right]$ represents the set of positions $x_1, x_2, \ldots, x_k$ of $k$ spatial lattice points connected to the particular auxiliary field involved for the $i$-th step of the update. $P_{k\times N}\left[x^{(i)}\right]$ is an index matrix labeling those positions with a matrix with $k$ rows and $N$ columns, in particular with the following form:

$$P_{k\times N}\left[x^{(i)}\right] = [e_{x_1}|e_{x_2}|\cdots|e_{x_k}]^T. \tag{22}$$

Additionally, $P_{N\times k}\left[x^{(i)}\right]$ is the transpose matrix of $P_{k\times N}\left[x^{(i)}\right]$:

$$P_{N\times k}\left[x^{(i)}\right] = \left(P_{k\times N}\left[x^{(i)}\right]\right)^T. \tag{23}$$

By repeatedly applying the recursive relation Eq. (21), one can obtain

$$A^{(i)} = \tilde{A}^{(i)} - X^{(i)}Y^{(i)}, \tag{24}$$

where $\tilde{A}^{(i)} = (I + X^{(i)}Y^{(i)})A^{(0)}$ with

$$X^{(i)} = P_{N \times ik}\left[x^{(1)}, \cdots, x^{(i)}\right]\begin{pmatrix} D^{(1)} & & \\ & \ddots & \\ & & D^{(i)} \end{pmatrix}, \tag{25}$$

$$Y^{(i)} = P_{ik \times N}\left[x^{(1)}, \cdots, x^{(i)}\right]. \tag{26}$$

We also define $\tilde{G}^{(i)} \equiv (\tilde{A}^{(i)})^{-1}$. In the simulation, we need $G^{(i)}$ instead of its inverse, and by using the Woodbury matrix identity and Eq. (24), we have

$$G^{(i)} = \left[(\tilde{G}^{(i)})^{-1} - X^{(i)}Y^{(i)}\right]^{-1} = \tilde{G}^{(i)} + \tilde{G}^{(i)}X^{(i)}\left[I_{ik \times ik} - Y^{(i)}\tilde{G}^{(i)}X^{(i)}\right]^{-1}Y^{(i)}\tilde{G}^{(i)}. \tag{27}$$

After simplification, the Green's function can be finally expressed in the following form

$$G^{(i)} = \left(G^{(0)} + G^{(0)}P_{N \times ik}\left[x^{(1)}, \cdots, x^{(i)}\right]\left(\Gamma^{(i)}_{ik \times ik}\right)^{-1} P_{ik \times N}\left[x^{(1)}, \cdots, x^{(i)}\right]G^{(0)}\right)E^{(i)}_{N \times N}. \tag{28}$$

This is the *key* formula of the submatrix update. Here we have introduced an $ik \times ik$ matrix $\Gamma^{(i)}_{ik \times ik}$ and a $N \times N$ matrix $E^{(i)}_{N \times N}$. They have the following forms:

$$\Gamma^{(i)}_{ik \times ik} \equiv \begin{pmatrix} I_{k \times k} + (D^{(1)})^{-1} & & \\ & \ddots & \\ & & I_{k \times k} + (D^{(i)})^{-1} \end{pmatrix} - P_{ik \times N}\left[x^{(1)}, \cdots, x^{(i)}\right]G^{(0)}P_{N \times ik}\left[x^{(1)}, \cdots, x^{(i)}\right], \tag{29}$$

$$E^{(i)}_{N \times N} \equiv I - P_{N \times ik}\left[x^{(1)}, \cdots, x^{(i)}\right]\begin{pmatrix} D^{(1)}(I_{k \times k} + D^{(1)})^{-1} & & \\ & \ddots & \\ & & D^{(i)}(I_{k \times k} + D^{(i)})^{-1} \end{pmatrix}P_{ik \times N}\left[x^{(1)}, \cdots, x^{(i)}\right].$$

After defining the matrix $\Gamma^{(i)}_{ik \times ik}$, we can discuss how to calculate the determinant ratio $\det\left[S^{(i)}\right]$ where

$$S^{(i)} = I_{k \times k} + \mathcal{V}^{(i)}D^{(i)}, \tag{30}$$

with

$$\mathcal{V}^{(i)} = -P_{k \times N}\left[x^{(i)}\right]\left(G^{(i-1)} - I\right)P_{N \times k}\left[x^{(i)}\right], \tag{31}$$

$$D^{(i)} = \begin{bmatrix} \Delta_{x_1^{(i)}x_1^{(i)}} & 0 & \cdots \\ 0 & \Delta_{x_2^{(i)}x_2^{(i)}} & \cdots \\ \vdots & \vdots & \ddots \end{bmatrix}_{k \times k}. \tag{32}$$

In order to calculate $P_{k \times N}\left[x^{(i)}\right]G^{(i-1)}P_{N \times k}\left[x^{(i)}\right]$ in Eq. (31), we can employ Eq. (28). It's evident that every time the determinant ratio is computed, the inverse of the $\Gamma^{(i)}_{ik \times ik}$ matrix is required. As the number of accepted configurations increases, the dimension of the $\Gamma^{(i)}_{ik \times ik}$ matrix also increases. Direct calculating the inverse of the $\Gamma^{(i)}_{ik \times ik}$ matrix after each acceptance would result in a computational complexity of $\mathcal{O}(i^3k^3)$ for each calculation, which is clearly not a good idea. To mitigate this issue, we use the block matrix inverse formula by rewriting $\Gamma^{(i)}_{ik \times ik}$ matrix (29) in a recursive form:

$$\Gamma^{(i)}_{ik \times ik} = \begin{pmatrix} \Gamma^{(i-1)} & W \\ Q & \Sigma \end{pmatrix}, \tag{33}$$

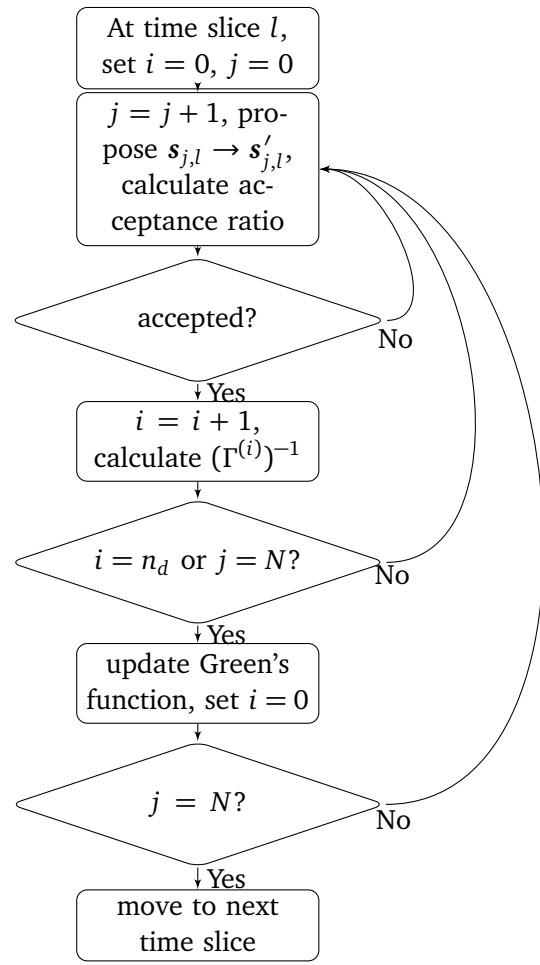

Figure 2: A flowchart of the submatrix update. The flowchart shows the detailed process within a time slice.

with

$$W = -P_{(i-1)k \times N}\left[x^{(1)}, \cdots, x^{(i-1)}\right] G^{(0)} P_{N \times k}\left[x^{(i)}\right], \tag{34}$$

$$Q = -P_{k \times N}\left[x^{(i)}\right] G^{(0)} P_{N \times (i-1)k}\left[x^{(1)}, \cdots, x^{(i-1)}\right], \tag{35}$$

$$\Sigma = I_{k \times k} + \left(D^{(i)}\right)^{-1} - P_{k \times N}\left[x^{(i)}\right] G^{(0)} P_{N \times k}\left[x^{(i)}\right]. \tag{36}$$

Using the following block matrix inverse formula

$$\begin{pmatrix} \Gamma & W \\ Q & \Sigma \end{pmatrix}^{-1} = \begin{pmatrix} \Gamma^{-1} + \Gamma^{-1} W \left(\Sigma - Q\Gamma^{-1}W\right)^{-1} Q\Gamma^{-1} & -\Gamma^{-1}W\left(\Sigma - Q\Gamma^{-1}W\right)^{-1} \\ -\left(\Sigma - Q\Gamma^{-1}W\right)^{-1}Q\Gamma^{-1} & \left(\Sigma - Q\Gamma^{-1}W\right)^{-1} \end{pmatrix}, \tag{37}$$

we can calculate the inverse of $\Gamma^{(i)}_{ik \times ik}$ matrix with computation complexity $\mathcal{O}(i^2 k^2)$. Eq. (28), Eq. (37) and intermediate matrices in Eqs. (30)-(32) and Eqs. (34)-(36) are foundations of submatrix update. In order to help readers to better understand how to use submatrix updates for local updates, we have created a flowchart illustrating the submatrix update process in Fig. 2.

Here, we offer a brief discussion on the computational complexity associated with submatrix update. The additional overhead for preparing the intermediate matrices primarily arises from computing $(\Gamma^{(i)})^{-1}$ instead of part of the Green's function in the delay update. Consequently, submatrix update reduces the computational complexity for the additional overhead from $\mathcal{O}(n_d^2 N)$ to $\mathcal{O}(n_d^3)$ per step of entire Green's function update. In each sweep, we perform a number of entire Green's function updates on the order of $\mathcal{O}(\beta N/n_d)$. This adjustment reduces the computational complexity of the additional overhead from $\mathcal{O}(\beta n_d N^2)$ in the delay update to $\mathcal{O}(\beta n_d^2 N)$ in the submatrix update. However, this reduction comes with a trade-off. In updating the Green's function, additional computations are necessary, that is $\left(\Gamma_{ik \times ik}^{(i)}\right)^{-1} P_{ik \times N}\left[x^{(1)}, \cdots, x^{(i)}\right] G^{(0)}$, which has complexity $\mathcal{O}(n_d^2 N)$. Fortunately, this can be done by Level 3 BLAS. Considering the factor of $\mathcal{O}(\beta N/n_d)$ number of entire Green's function updates per sweep, this additional calculation has complexity $\mathcal{O}(\beta n_d N^2)$.

To facilitate readers in comparing the computational complexity and the corresponding types of computations between different updating methods, we summarize the results in Table. 1. Clearly, compared to the delay update, submatrix update reduces the number of Level 1 BLAS calculations, with a trade-off to have more Level 3 BLAS computations which has higher efficiency.

Furthermore, we note that the optimized value of $n_d$ is limited by the size of the cache. Based on Ref. [1] and our tests on various platforms, setting $kn_d$ to $2^\lambda$, where $\lambda = [\log_2 \frac{N}{12}]$, has proven to be a good choice. For example, for $N = 60^2$, $kn_d$ can be set to 256. The above discussion is based on the finite temperature DQMC, and the zero-temperature version is presented in the Appendix D.

## 3 Model and results

To contrast the computational efficiency and the optimization capability in multi-threaded computations of the submatrix update algorithm and the delay update algorithm, let's first consider onsite interactions ($k = 1$). Note that a case with extended interaction ($k = 2$) is also considered in Appendix C. We first consider the Hubbard model on a square lattice, the Hamiltonian is:

$$H_{tU} = -t \sum_{\langle i,j \rangle, \alpha} (c_{i,\alpha}^\dagger c_{j,\alpha} + \text{H.c.}) + \frac{U}{2} \sum_i (n_i - 1)^2, \tag{38}$$

where $c_{i,\alpha}^\dagger$ represents the creation operator for an electron on site $i$ with spin polarization $\alpha = \uparrow / \downarrow$. The operator $n_i = \sum_\alpha c_{i\alpha}^\dagger c_{i\alpha}$ denotes the fermion number density on site $i$. The parameter $t$ describes the hopping of electrons between NN sites $i$ and $j$, while $U$ represents the amplitude of the onsite Hubbard repulsive interaction. We set $t = 1$ as the unit of energy and focus on the half-filling case to avoid the sign problem.

After the Hubbard-Stratonovich (HS) transformation, the spin up and down sectors are block diagonalized, allowing for separate calculations of the Green's function for each spin polarization. This is why the Hubbard model serves as an example of the $k = 1$ case for local updates. When $U/t = 0$, the model at half-filling exhibits a diamond-shaped Fermi surface. However, turning on $U/t$ triggers a metal-insulator transition to an antiferromagnetic insulator phase.

To maintain controlled variables, we set the Hubbard repulsive interaction $U/t = 1$, the size of the system $L = 60$, the inverse temperature $\beta t = 1$, and the time slice for Trotter decomposition $a_\tau t = 0.1$, ensuring consistent initial conditions for both algorithms. Each simulation comprises 8 Markov chains, with each chain undergoing 25 sweeps. Let's consider finite-temperature calculations. We record the average computation time for updates

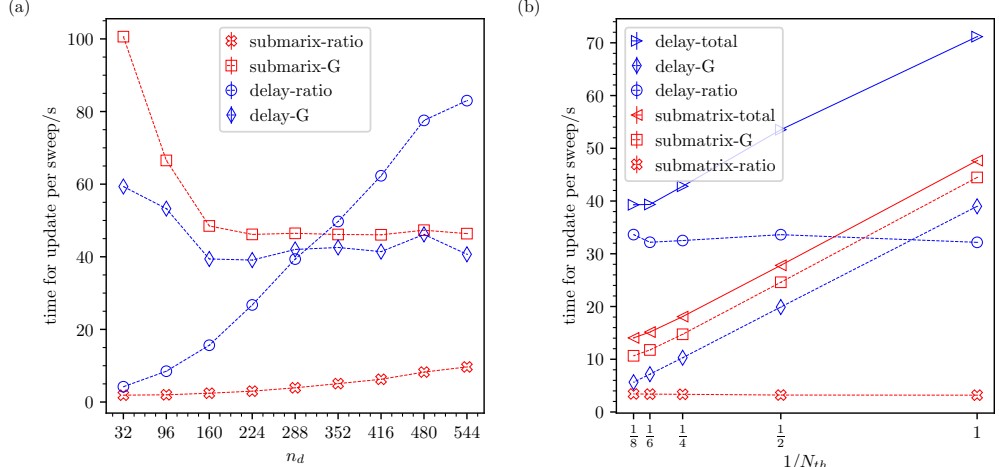

Figure 3: (a) The time for update per sweep taken by delay update and submatrix update in DQMC-finite-T to compute the acceptance ratio and intermediate matrices (-ratio) and update the Green's function matrix (-G) of the Hubbard model on a square lattice at different $n_d$ when the number of threads $N_{th}$ is 1. (b) The time for update per sweep taken by delay update and submatrix update in DQMC-finite-T to compute the acceptance ratio and intermediate matrices(-ratio), update the Green's function matrix (-G) and the total update time (-total) of the Hubbard model on a square lattice at different $N_{th}$ when $n_d = 256$.

per sweep using different local update algorithms for comparison. To study the computational complexity of both algorithms at different $n_d$, we recorded the computation time for delayed update and submatrix update at various $n_d$. For ease of comparison, we divided the computation time into two parts: one for computing the acceptance ratio and intermediate matrices, denoted as '-ratio,' and the other for updating the Green's function matrix, denoted as '-G.' Additionally, we recorded the total time used for the updating process, defined as '-total.' The results are shown in Fig. 3(a).

From the results, it is clear that the time taken to compute the acceptance ratio and intermediate matrices in submatrix update is significantly less than that in delay update. However, the time taken to update the Green's function matrix is slightly longer in submatrix update compared to delay update, which is consistent with the earlier analysis of their computational complexity. Regardless of whether it's delay update or submatrix update, as $n_d$ increases, the time taken to compute the acceptance ratio increases accordingly, while the time taken to update the Green's function decreases and eventually converges. Therefore, both methods have an optimal $n_d$, as mentioned earlier.

To investigate the optimization of different computational components during updating by multiple threads, we fixed $n_d$ for both algorithms to 256 and varied the number of threads $N_{th}$ as 1, 2, 4, 6 and 8. We recorded the time taken for each component under different number of threads. The results are shown in the Fig. 3(b). When the number of threads increases, the time taken to compute the acceptance ratio remains almost constant, or even slightly increases. This is because the ratio calculation part is fragmented and cannot be efficiently paralleled. However, when updating the Green's function matrix, which uses Level 3 BLAS and can be effectively paralleled.

It is also interesting to point out that the speedup of multiple threads running is limited by $n_d$, as we increase the number of threads, the speedup has a tendency to saturate. To verify this effect, we perform tests on different $n_d$ and different numbers of threads $N_{th}$, and summarize

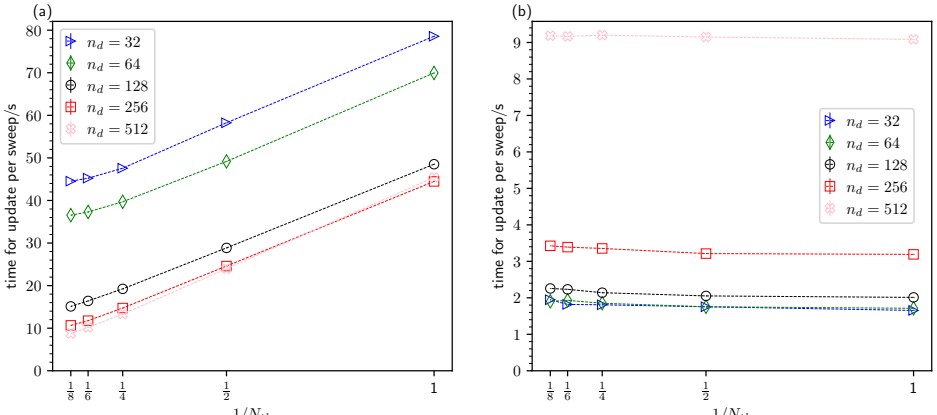

Figure 4: (a) The relationship between the time for update per sweep taken to update the Green's function matrix in *submatrix update* and the number of threads $N_{th}$ for different $n_d$ values in DQMC-finite-T of the Hubbard model on a square lattice $L = 60$. (b) The relationship between the time for update per sweep taken to calculate the acceptance ratio and intermediate matrices in *submatrix update* and the number of threads $N_{th}$ for different $n_d$ values in DQMC-finite-T of the Hubbard model on a square lattice $L = 60$.

the results in the Fig. 4. It is obvious that as $n_d$ increases, the part of updating the Green's function matrix can utilize threads more efficiently. However, the increase in $n_d$ leads to an increase in the time taken for computing the acceptance ratio, which cannot be accelerated by multiple threads. Therefore, it is necessary to choose an appropriate $n_d$ to maximize the speedup. Similar test for delay update will be discussed in the Appendix B. According to the previous discussion and Ref. [68], submatrix update has a larger optimal $n_d$ and less Level 1 BLAS calculations than delay update. Therefore, submatrix update is not only more efficient in single-thread running, but is also more suitable for multi-threaded computations. This is crucial for using large-scale parallel computations.

To further demonstrate the capability of submatrix update, we utilize this method to compute the phase diagram of the Néel transition in the half-filled Hubbard model on a 3D cubic lattice. In the literatures, only system sizes up to 1000 lattice sites ($N = 1000$) [51], are available. We show that with the help of submatrix update, it becomes feasible to compute much larger system sizes for the 3D Hubbard model.

When the 3D Hubbard model enters the antiferromagnetic phase, it develops long range spin correlations. The Fourier transform of the spin correlation function defines the spin structure factor, which has following form,

$$S(\mathbf{q}) \equiv \frac{1}{N} \sum_{i,j} e^{i\mathbf{q} \cdot (\mathbf{r}_i - \mathbf{r}_j)} \langle \vec{S}_i \cdot \vec{S}_j \rangle . \tag{39}$$

The $\mathbf{q} = \mathbf{Q} \equiv (\pi, \pi, \pi)$ structure factor $S(\mathbf{Q})$ is related to the magnetization $m$. In the thermodynamic limit (TDL), $\lim_{L \to \infty} S(\mathbf{Q})/N = m^2$. To study the Néel phase transition, we introduce a correlation ratio, which is a dimensionless quantity with following form:

$$r_S \equiv 1 - \frac{S(\mathbf{Q} + d\mathbf{q})}{S(\mathbf{Q})}, \tag{40}$$

where $d\mathbf{q} = (\frac{2\pi}{L}, 0, 0)$. The correlation ratio can be used to identify the critical point of the phase transition. As the finite temperature Néel transition is known to be a $O(3)$ transition,

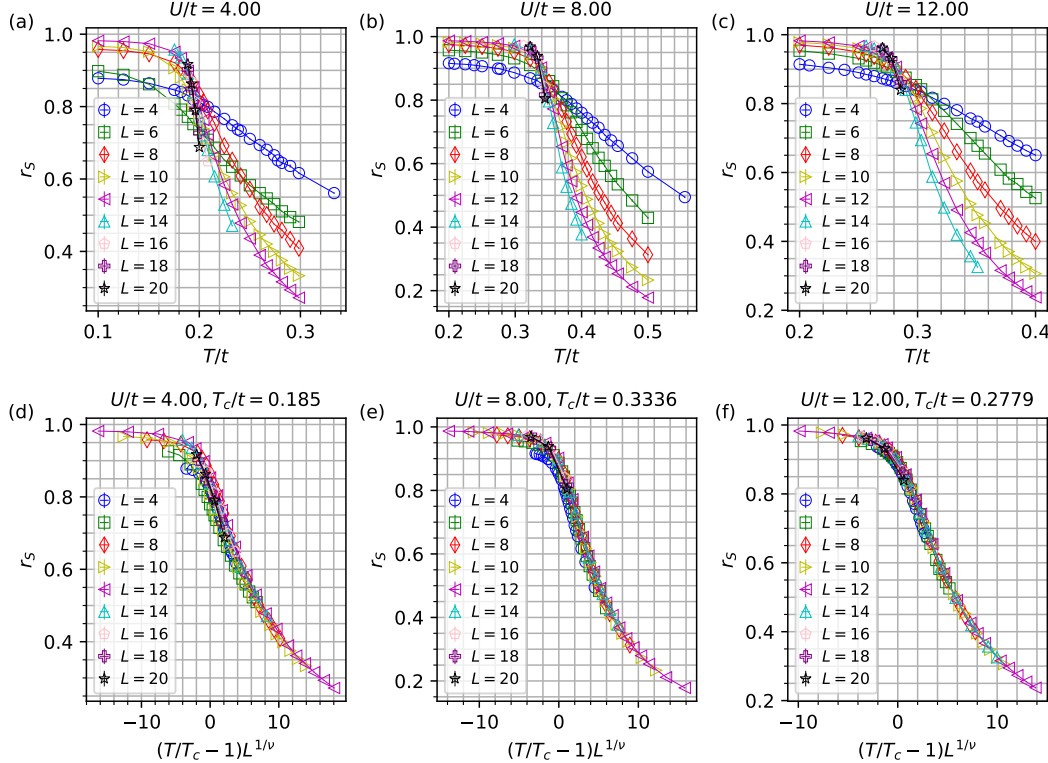

Figure 5: (a), (b) and (c), scanning of the correlation ratio $r_S$ at different temperatures for $U/t = 4.00,\ 8.00,\ 12.00$ to obtain the critical temperatures $T_c$. (d), (e) and (f), the data collapse of the $r_S$ for $U/t = 4.00,\ 8.00,\ 12.00$, where $\nu = 0.707$.

its critical exponents is well known, which can be used to double check our calculations. The correlation ratio as a dimensionless quantity has following scaling behavior

$$r_S(T, L) = f\left(\frac{T - T_c}{T_c} L^{\frac{1}{\nu}}\right),\tag{41}$$

where $\nu \approx 0.707$. At the transition temperature $T_c$, the $r_S - T$ curves for different $L$ intersects. We consider $U/t = 4, 6, 8, 10, 12$ and perform scans of $r_S$ of the 3D Hubbard model on a cubic lattice ($N = L^3$) at different temperatures using submatrix update for various $L$ values (up to $L = 20$) to get the critical temperature $T_c$. The results are shown in Fig. 5 and 7.

From the computational results, as mentioned earlier, below the Néel temperature, the system enters the Néel state, where the spin structure factor $S(\mathbf{Q})$ tends to diverge. This results in $r_S$ increasing from 0 to 1 after entering the Néel state. According to previous analysis, $r_S$ for different sizes eventually intersect at a single point, and the abscissa of this point corresponds to the critical temperature at the Néel phase transition. However, the actual results may deviate from this prediction due to finite-size effects. To ensure the reliability of the results, we need to analyze the finite-size effects and extrapolate them to the TDL. We search for the intersections of $r_S$ between adjacent sizes ($L$ and $L + 2$), extract these intersection points, and extrapolate them to $L \to \infty$ using function $T = aL^{-b} + T_c$ to find the critical temperature in the TDL. The results are shown in Fig. 6. From the results, we successfully obtained the Néel transition temperatures in the thermodynamic limit using this method for $U/t = 6, 8, 10$ and 12. However, for $U/t = 4$, the intersection points fluctuate, making it difficult to extrapolate using adjacent points. This is due to the stronger finite-size effects when $U$ is small. Therefore, we divided the points into two groups using next adjacent size ($L$ and $L + 4$), and

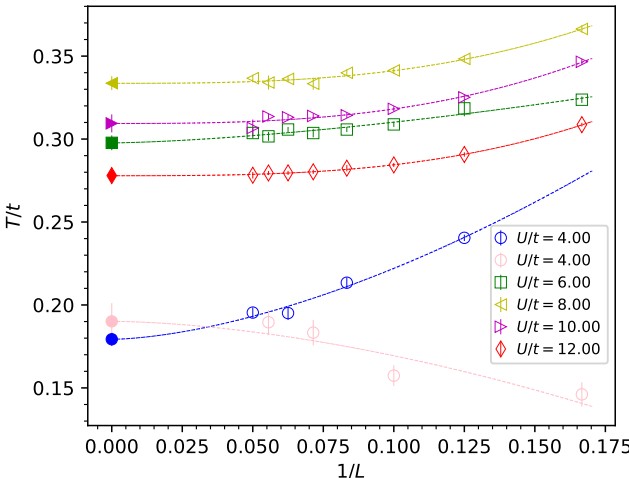

Figure 6: The finite-size effects for different $U/t$ values are analyzed using correlation ratio crossing. The uncertainty is estimated by moving the intersection segment within the error bars of $r_S$ between adjacent sizes ($L$ and $L + 2$) for $U/t = 6.00, 8.00, 10.00, 12.00$ and next adjacent size ($L$ and $L + 4$) for $U/t = 4.00$. Solid symbols are used to represent extrapolated results.

utilize the function $T = aL^{-b} + T_c$ for separate extrapolation for each group. The critical transition temperatures obtained by extrapolating these two sets of data are $0.190 \pm 0.013$ and $0.1793 \pm 0.0039$. We found that both results fall within the error bar. Ultimately, we chose their average value as the final value for the critical transition temperature at $U/t = 4$. The final result for the critical Néel transition temperature of the 3D half-filled Hubbard model is shown in Table 2.

We utilize this data to plot the phase diagram of the Néel transition for the 3D half-filled Hubbard model on a cubic lattice, as shown in Fig. 1(b).

Finally, we validate the accuracy of the critical transition temperatures obtained for different interaction strengths. According to the description of formula Eqs. (41) earlier, when the correct critical temperature is obtained, data from all different sizes will collapse onto a single curve. The final result is shown in Fig. 5 and 7.

It shows that when $U/t$ is relatively small, the finite-size effects from small sizes do not result in a good collapse onto a single curve. We will explain in the Appendix G why size effects are more pronounced under weak coupling conditions. However, as the size or $U/t$ increases, the influence of finite-size effects decreases significantly, eventually allowing the data to collapse onto a single curve. This also indicates the necessity of computing large-size models to obtain more accurate results.

Table 2: The Néel temperature of the 3D half-filled Hubbard model on a cubic lattice.

| U/t | $T_c$/t |
|-----|---------|
| 4 | 0.185(14) |
| 6 | 0.2977(45) |
| 8 | 0.3336(42) |
| 10 | 0.3094(54) |
| 12 | 0.2779(20) |

## 4 Conclusion and discussion

In this work, we have developed a general submatrix update method to further optimize the computational efficiency of DQMC. This method is applicable to both onsite and extended interactions and demonstrates optimization capabilities under both zero-temperature and finite-temperature conditions. In our tests, the submatrix update method not only exhibits higher computational efficiency compared to delay update for single thread running, but also offers better optimization in multi-threaded computations, which is more suitable for large size calculations. Subsequently, we utilized this method to compute the phase diagram of the Néel transition for the 3D half-filled Hubbard model on a cubic lattice. By computing system sizes up to 8,000 sites, we obtained more accurate phase diagram of 3D Hubbard model at half-filling. This result will be useful for guiding the cold atom simulation of 3D Hubbard model [71].

Looking forward, our submatrix update is general, and can be used to accelerate the simulation of 2D Dirac fermions with interactions. By pushing the simulation of the system size to the order of $100^2$, we foresee that we will have converging results for the critical exponents of Gross-Neveu transitions in many interacting Dirac fermion systems. At the same time, it will help to distinguish from weakly first order to a continuous transition in the situations when the finite size effect is very large. It will brings more insights into possible deconfined quantum criticality in fermionic systems, as well as shed light on critical Fermi surface problem where simulating larger system size is essential.

*Note added* — After the submission of our manuscript to arXiv, we became aware of another work [79] that was concurrently posted, which explores simulations of the 3D Hubbard model with delay update.

## Acknowledgments

**Funding information** X.Y.X. is sponsored by the National Key R&D Program of China (Grant No. 2022YFA1402702, No. 2021YFA1401400), the National Natural Science Foundation of China (Grants No. 12274289, No. 12447103), the Innovation Program for Quantum Science and Technology (under Grant no. 2021ZD0301902), Yangyang Development Fund, and startup funds from SJTU. The computations in this paper were run on the Siyuan-1 and $\pi$ 2.0 clusters supported by the Center for High Performance Computing at Shanghai Jiao Tong University.

## A  Other data for 3D Hubbard model

The correlation ratio and its data collapse for $U/t = 6.00$ and $U/t = 10.00$ are presented in Fig. 7.

## B  Delay update under multi-threads

To compare with the submatrix update results under multi-threads as shown in Fig. 4, we also perform similar tests for the delay update. We set $n_d$ for delay update to 32, 64, 128, 256 and 512, and consider different numbers of threads $N_{th}$. We record the time taken for each component, and the results are similar to submatrix update as shown in the Fig. 8. The Green's function update part shows nearly perfect speedup under multi-threads running, while the ratio calculation part does not show any significant speedup. Therefore, reducing the time cost for the ratio calculation part is essential to improve the efficiency, and that is exactly what submatrix update algorithm do.

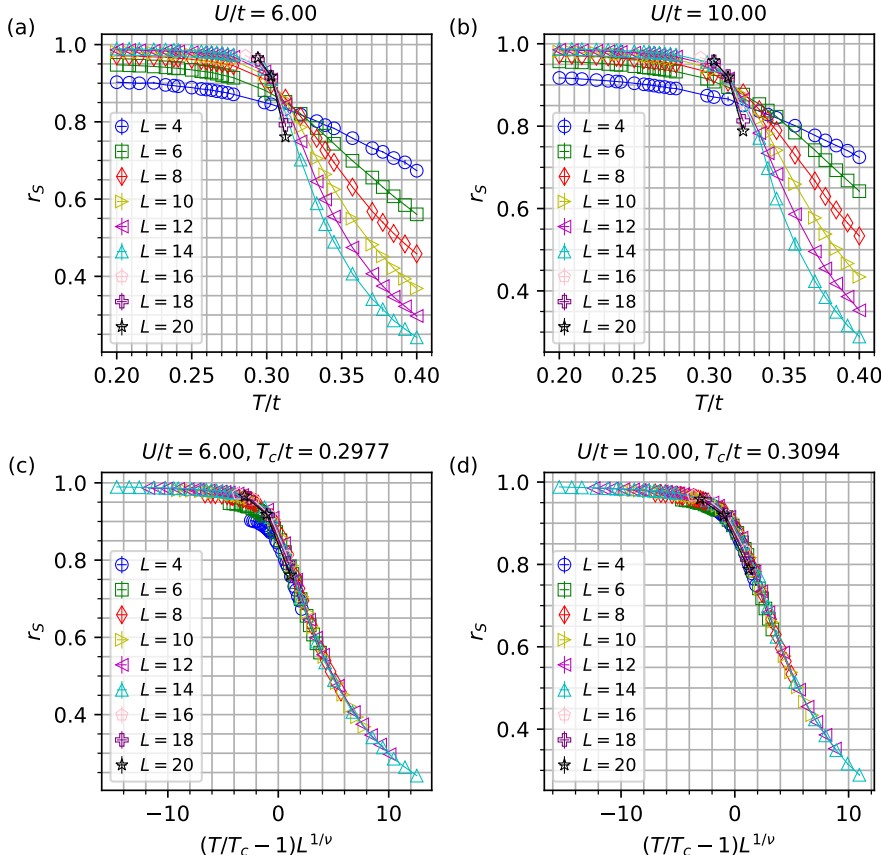

Figure 7: (a) and (b), scanning of $r_S$ at different temperatures for $U/t = 6.00$, 10.00 to obtain the critical temperatures $T_c$. (c) and (d), the data collapse of the $r_S$ for $U/t = 6.00$, 10.00, where $\nu = 0.707$.

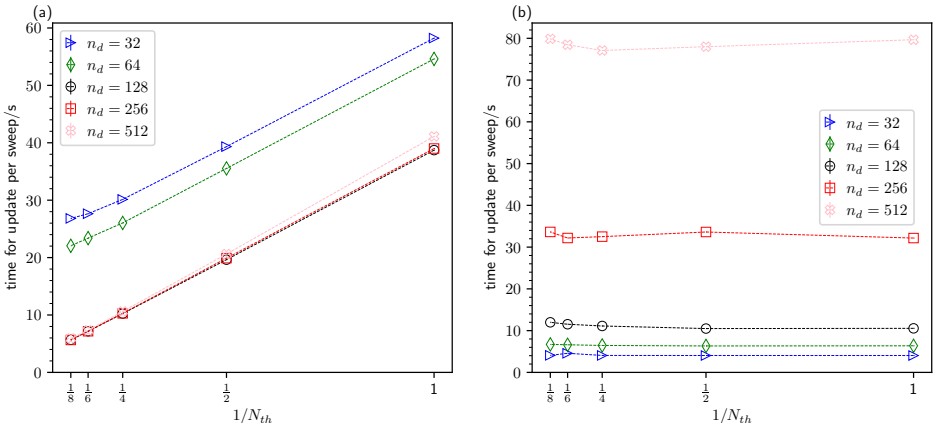

Figure 8: (a) The relationship between the time for update per sweep taken to update the Green's function matrix in *delay update* and the number of threads $N_{th}$ for different $n_d$ values in DQMC-finite-T of the Hubbard model on a square lattice $L = 60$. (b) The relationship between the time for update per sweep taken to calculate the acceptance ratio and intermediate matrices in *delay update* and the number of threads $N_{th}$ for different $n_d$ values in DQMC-finite-T of the Hubbard model on a square lattice $L = 60$.

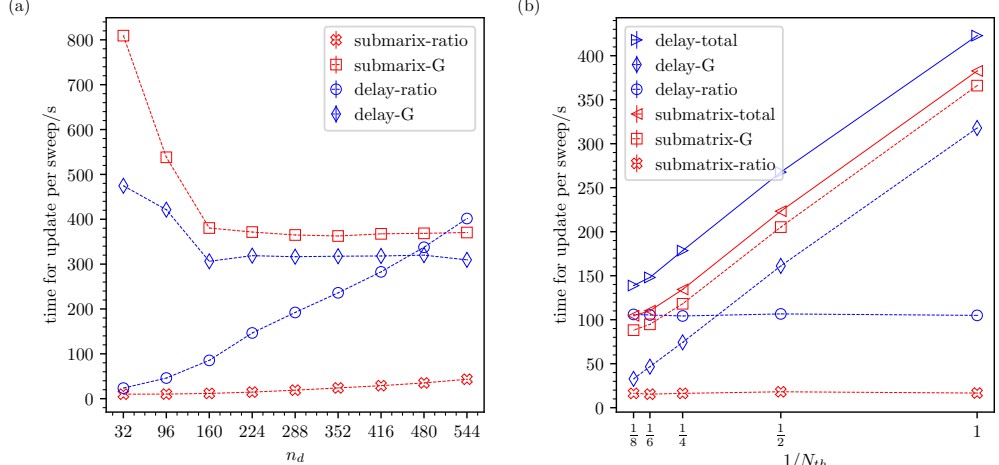

Figure 9: (a) The time for update per sweep taken by delay update and submatrix update in DQMC-finite-T to compute the acceptance ratio (-ratio) and update the Green's function matrix (-G) of the $t$-$V$ model on a square lattice at different $n_d$ when the number of threads $N_{th}$ is 1. (b) The time for update per sweep taken by delay update and submatrix update in DQMC-finite-T to compute the acceptance ratio and intermediate matrices (-ratio), update the Green's function matrix (-G) and the total update time (-total) of the $t$-$V$ model on a square lattice at different $N_{th}$ when $n_d = 256$.

## C  The submatrix update of DQMC-finite-T in spinless t-V model

We have demonstrated the optimization achieved by submatrix update in the presence of onsite interaction ($k = 1$). Now, let's explore its optimization in extended interaction. We consider a model where $\Delta$ contains only two non-zero diagonal elements ($k = 2$). The $t$-$V$ model serves as a such kind of example. It is a spinless fermion model with NN interaction. Let's consider this model on a square lattice. The Hamiltonian is written as

$$H_{tV} = -t \sum_{\langle i,j \rangle} (c_i^\dagger c_j + \text{H.c.}) + V \sum_{\langle i,j \rangle} \left( n_i - \frac{1}{2} \right) \left( n_j - \frac{1}{2} \right). \tag{C.1}$$

In the model, $c_i^\dagger$ represents the fermionic creation operator on site $i$, $t$ denotes the NN hopping amplitude, and $V > 0$ signifies the density repulsive interaction between NN sites. We maintain $t = 1$ as the unit of energy and concentrate on the half-filling case, which still allows for a suitable Hubbard-Stratonovich transformation to circumvent the sign problem [29, 30]. However, this transformation leads to two non-zero off-diagonal elements in $\Delta$ during updates. To enable the use of submatrix updates, we require $\Delta$ to have only non-zero diagonal elements. We achieve this by diagonalizing $e^{V(s_{i,l})}$ and propagating the Green's function to obtain a representation where $\Delta$ is diagonal and contains only two non-zero diagonal elements.

In the simulation, we set $V/t = 1$, and keep all other conditions identical to those of the Hubbard model discussed earlier. Subsequently, we separately employed delay update and submatrix update to conduct DQMC-finite-T calculations on the spinless $t$-$V$ model. We then compared the performance of these two update methods. It exhibits almost similar phenomena as the Hubbard model, as shown in Fig. 9.

Table 3: The computational complexity and types of computations required for various local updates in DQMC-zero-T. Here, 'update-ratio' refers to the calculations needed to obtain intermediate matrices/vectors for calculating the determinant ratio and to accumulate vectors used to finally update the Green's function, and 'update-G' refers to the calculations for updating the entire Green's function. 'Level 1' means Level 1 BLAS, and 'Level 3' means Level 3 BLAS.

| | fast update | delay update | submatrix update |
|---|---|---|---|
| update-ratio | - | $\mathcal{O}(\beta n_d N^2)$ | $\mathcal{O}(\beta n_d^2 N)$ |
| | | Level 1 | Level 1 |
| update-$G$ | $\mathcal{O}(\beta N_p^2 N)$ | $\mathcal{O}(\beta N^3)$ | $\mathcal{O}(\beta N^3 + \beta n_d N^2)$ |
| | Level 1 | Level 3 | Level 3 |

## D  The submatrix update in the zero-temperature version of DQMC

The aforementioned submatrix update scheme can also be applied to the zero-temperature version of DQMC (DQMC-zero-T). In DQMC-zero-T, the normalization factor of the ground state wavefunction $|\Psi_0\rangle$ plays a role similar to the partition function in the finite-temperature case. We perform the HS transformation on the interaction part and trace out the fermions' degree of freedom in the Fock space with a fixed number of particles $N_p$. Then, we have:

$$\langle \Psi_0 | \Psi_0 \rangle = \sum_s \det\left[ P^\dagger B_s(2\Theta, 0) P \right], \tag{D.1}$$

where we have represent the ground state wavefunction as a projection of a certain trial wavefunction, that is $|\Psi_0\rangle = e^{-\Theta H}|\Psi_T\rangle$, where $\Theta$ is the projection time and needs to be sufficiently large to project to the ground state. The trial wavefunction can be represented as a Slater determinant $|\Psi_T\rangle = \prod_{i=1}^{N_p}(\mathbf{c}^\dagger P)_i|0\rangle$, where $|0\rangle$ is a vacuum state, $N_p$ is the number of particles in the ground state, and $P$ is a matrix with dimensions $N \times N_p$. Typically, we choose the trial wavefunction to be the ground state of the non-interacting part $H_0 = \mathbf{c}^\dagger K \mathbf{c}$, and then $P$ consists of $N_p$ lowest eigenvectors of $K$. For consistency, we will interchangeably use $2\Theta$ and $\beta$ when necessary.

In the zero-temperature case, we define $B^\rangle(\tau) \equiv B(\tau, 0)P$ and $B^\langle(\tau) \equiv P^\dagger B(2\Theta, \tau)$. For the simplification, we will omit $\tau$ dependence of $B^\langle$ and $B^\rangle$ in the following discussion by default. In DQMC-zero-T, during the fast update, we keep track of $B^\langle$, $B^\rangle$, and $\left(B^\langle B^\rangle\right)^{-1}$ instead of the Green's function. After each local update, $B^\rangle$ is updated to $(I+\Delta)B^\rangle$, and the formula to calculate the determinant ratio is identical to the finite temperature case as shown in Eq. (9). The only extra calculation is that the Green's function elements in $\mathcal{V}$ as shown in Eq. (11) should be calculated explicitly. Note that Green's function $G(\tau, \tau) = I - B^\rangle \left(B^\langle B^\rangle\right)^{-1} B^\langle$ in DQMC-zero-T, so the calculation of Green's function elements for calculating ratio has complexity $\mathcal{O}(\beta N N_p^2)$ per sweep. If the update is accepted, we update $B^\rangle$ and $\left(B^\langle B^\rangle\right)^{-1}$, and the computational complexity is also $\mathcal{O}(\beta N N_p^2)$ per sweep. According to the previous derivation, we can apply the submatrix update to the zero-temperature case by working with the so called $F^{(i)}$ matrix instead of working with $B^\langle$, $B^\rangle$, and $\left(B^\langle B^\rangle\right)^{-1}$. In DQMC-zero-T, $F^{(i)}$ can be defined as:

$$
\begin{aligned}
F^{(i)} &\equiv \left(B^\rangle\right)^{(0)} \left(\left(B^\langle\right)^{(i)}\left(B^\rangle\right)^{(i)}\right)^{-1} \left(B^\langle\right)^{(0)} \\
&= F^{(0)} - F^{(0)} P_{N\times ik}\left[x^{(1)}, \cdots, x^{(i)}\right]\left(\Gamma_{ik\times ik}^{(i)}\right)^{-1} P_{ik\times N}\left[x^{(1)}, \cdots, x^{(i)}\right] F^{(0)},
\end{aligned}
\tag{D.2}
$$

and

$$F^{(0)} = \left(B^\rangle\right)^{(0)} \left(\left(B^\langle\right)^{(0)} \left(B^\rangle\right)^{(0)}\right)^{-1} \left(B^\langle\right)^{(0)} . \tag{D.3}$$

Note that $F^{(i)}$ plays the same role in submatrix update in DQMC-zero-T as that of Green's function $G^{(i)}$ in submatrix update in DQMC-finite-T. Therefore, Eq. (D.2) is the *key* formula for submatrix update in DQMC-zero-T. To obtain above formula, we have utilized

$$\left(B^\rangle\right)^{(i)} = \left(I + X^{(i)} Y^{(i)}\right) \left(B^\rangle\right)^{(0)} , \tag{D.4}$$

$$\left(B^\langle\right)^{(i)} = \left(B^\langle\right)^{(0)} , \tag{D.5}$$

where $X^{(i)}$ and $Y^{(i)}$ are defined in Eq. (25) and Eq. (26). Also note that following relations

$$P_{k \times N}[x^{(i+1)}] \left(B^\rangle\right)^{(i)} = P_{k \times N}[x^{(i+1)}] \left(B^\rangle\right)^{(0)} , \tag{D.6}$$

$$\left(B^\langle\right)^{(i)} P_{N \times k}[x^{(i+1)}] = \left(B^\langle\right)^{(0)} P_{N \times k}[x^{(i+1)}], \tag{D.7}$$

are useful for obtaining Green's function elements for the determinant ratio calculation in particularly constructing $\mathcal{V}$ defined in Eq. (11). For example, in the $i$-th step, $\mathcal{V}^{(i)}$ can be calculated by

$$\begin{aligned} \mathcal{V}^{(i)} &= P_{k \times N}[x^{(i)}] \left(I - G^{(i-1)}\right) P_{N \times k}[x^{(i)}] \\ &= P_{k \times N}[x^{(i)}] F^{(i-1)} P_{N \times k}[x^{(i)}]. \end{aligned} \tag{D.8}$$

In the practical calculation, $F^{(i)}$ is not calculated at any intermediate step, instead, we keep track of $\left(\Gamma^{(i)}\right)^{-1}$ as in the submatrix update for DQMC-finite-T. We only perform the update of $F^{(i)}$ when $i = n_d$. This step is also conveniently referred to as Green's function update as it is quite similar to the entire Green's function update in DQMC-finite-T. One can see, the submatrix update formula for zero-temperature case is quite similar to finite-temperature case with only a little difference. At the beginning of local update in each time slice, we have to calculate $F^{(0)}$ using Eq. (D.3) which has computational complexity $\mathcal{O}(N_p^2 N + N_p^3)$ in each time slice, and we have $\beta$ time slices, therefore the computational complexity is $\mathcal{O}(\beta N_p^2 N + \beta N_p^3)$ per sweep. Note that computational complexity of submatrix update based on $F^{(i)}$ is $\mathcal{O}(\beta N^3 + \beta n_d N^2)$, similar to the DQMC-finite-T case. As $N_p < N$, the computational complexity for the Green's function update in total is still $\mathcal{O}(\beta N^3 + \beta n_d N^2)$ in submatrix update. Since the calculation is dominant by the calculation with complexity $\mathcal{O}(\beta N^3)$, the time taken for updating the Green's function matrix in DQMC-zero-T is almost the same for both delay update and submatrix update, as shown in Fig. 10 for Hubbard model and Fig. 11 for $t$-$V$ model. Again, submatrix update has better performance both for single thread running and multi-thread running as it significantly reduce the Level 1 BLAS calculation during obtaining acceptance ratio and intermediate matrices.

# E  Data collapse of the magnetization $m$

To further verify the accuracy of the phase diagram, we use the fact that the Néel transition is an $O(3)$ phase transition, with known critical exponents, to cross-check other critical exponents. We choose the magnetization $m$ for this analysis, which has the following relationship with system size and critical temperature:

$$m^2(T, L) L^{\frac{2\beta}{\nu}} = f'\left(\frac{T - T_c}{T_c} L^{\frac{1}{\nu}}\right), \tag{E.1}$$

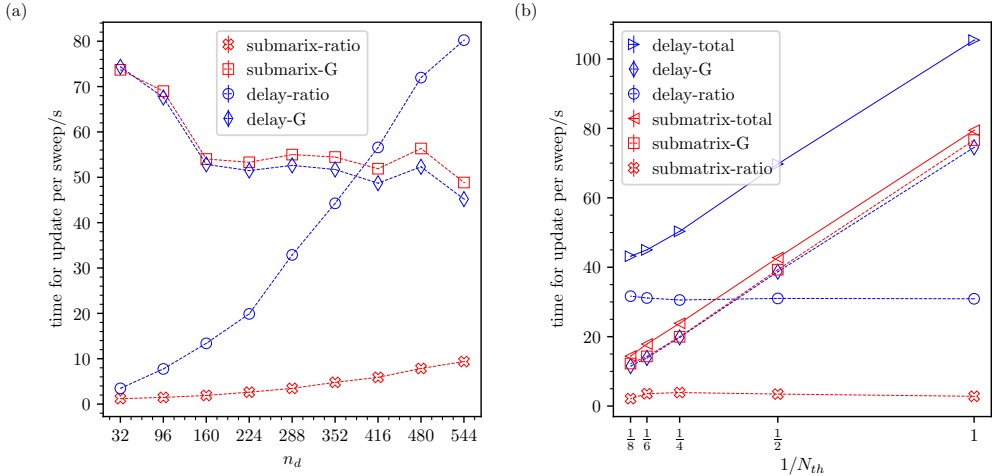

Figure 10: (a) The time for update per sweep taken by delay update and submatrix update in DQMC-zero-T to compute the acceptance ratio and intermediate matrices (-ratio) and update the Green's function matrix (-G) of the Hubbard model on a square lattice at different $n_d$ when the number of threads $N_{th}$ is 1. (b) The time for update per sweep taken by delay update and submatrix update in DQMC-zero-T to compute the acceptance ratio and intermediate matrices (-ratio), update the Green's function matrix (-G) and the total update time (-total) of the Hubbard model on a square lattice at different $N_{th}$ when $n_d = 256$.

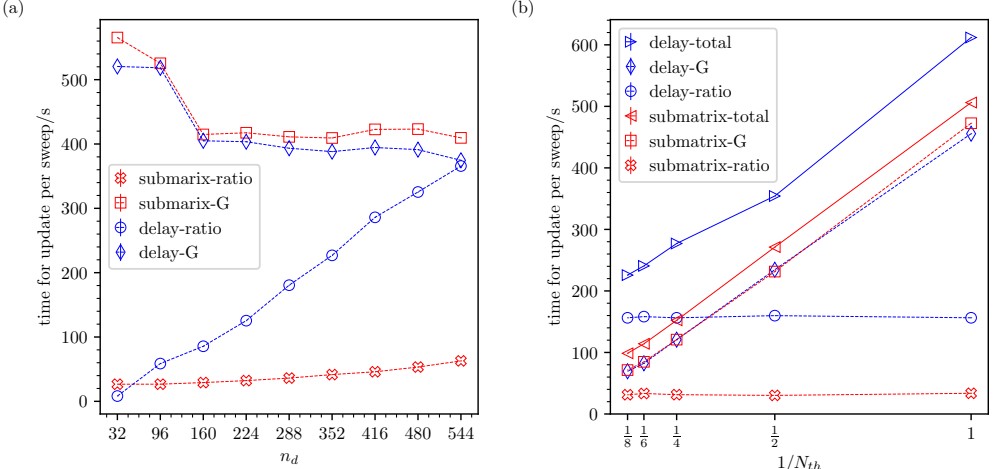

Figure 11: (a) The time for update per sweep taken by delay update and submatrix update in DQMC-zero-T to compute the acceptance ratio and intermediate matrices (-ratio) and update the Green's function matrix (-G) of the $t$-$V$ model on a square lattice at different $n_d$ when the number of threads $N_{th}$ is 1. (b) The time for update per sweep taken by delay update and submatrix update in DQMC-zero-T to compute the acceptance ratio and intermediate matrices (-ratio), update the Green's function matrix (-G) and the total update time (-total) of the $t$-$V$ model on a square lattice at different $N_{th}$ when $n_d = 256$.

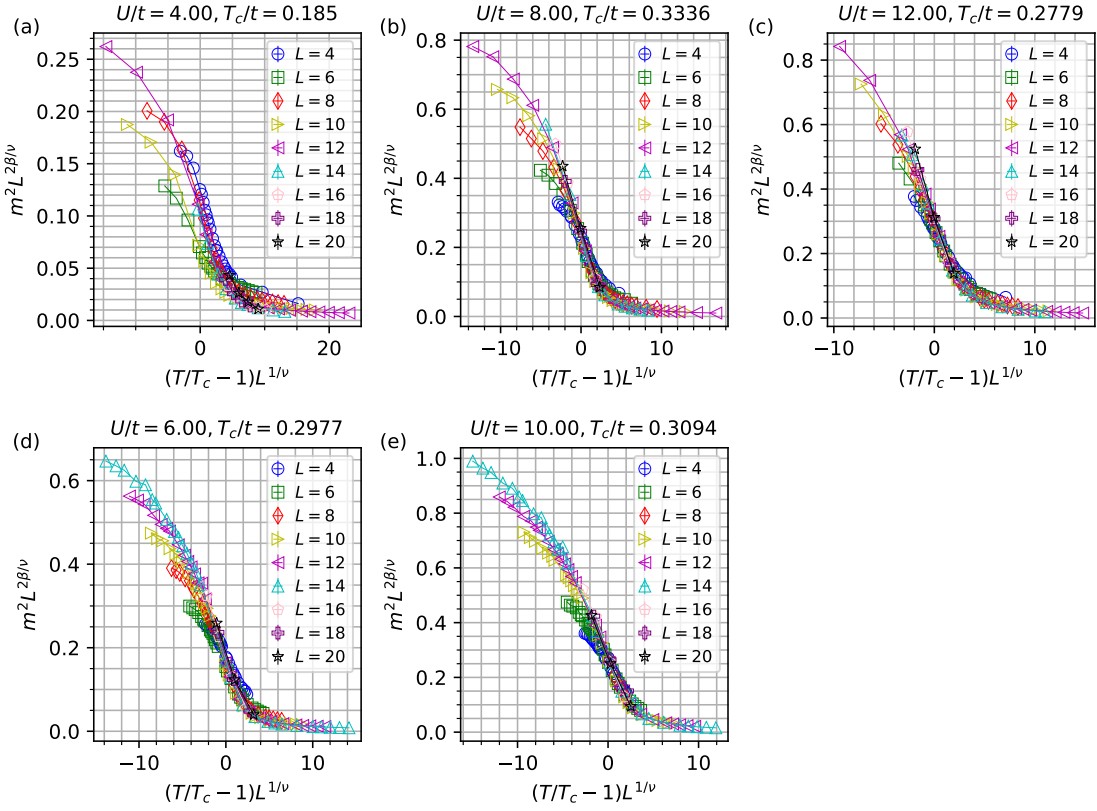

Figure 12: (a), (b), (c), (d) and (e), the data collapse of the $m^2(T, L)$ for $U/t = 4.00$, $8.00$, $12.00$, $6.00$, $10.00$, where $\nu = 0.707$ and $\beta = 0.366$.

where $\beta \approx 0.366$ and $\nu \approx 0.707$ are the critical exponents for magnetization and correlation length, respectively, and $f'$ is another scaling function. By fitting this relation, we can check the consistency of the obtained critical exponents with those of the $O(3)$ universality class. We still consider $U/t = 4, 6, 8, 10, 12$ and perform calculations for $m^2(T, L) = S(\mathbf{Q})/N$ at different temperatures and system size $L$ (up to $L = 20$) to validate the accuracy of the critical transition temperatures obtained for different interaction strengths. According to the description of formula Eqs. (E.1), when the correct critical temperature is obtained, data from all different sizes will collapse onto a single curve, as shown in Fig. 12.

It also shows that when $U/t$ is relatively small, the finite-size effects from small sizes do not result in a good collapse onto a single curve. However, as the size or $U/t$ increases, the influence of finite-size effects decreases significantly, eventually allowing the data to collapse onto a single curve as the same as the $r_S$.

# F  The choice of the $a_\tau$

Due to the mentioned system errors introduced by Trotter decomposition, we need to ensure that extrapolating the phase transition point is not affected by Trotter errors. This requires $a_\tau \to 0$. However, when $a_\tau$ is small, it leads to numerous time slices, consuming significant computational resources. Hence, we need to find an appropriate $a_\tau$. To determine the suitable $a_\tau$, we conducted a test before the computation. As $a_\tau$ decreases, $r_S$ continuously decreases, and eventually converges around $a_\tau t = 0.05$ as shown in Fig. 13. To further verify the re-

lationship between the choice of time slice and the system size, we performed calculations with different values of $a_\tau$ near the critical temperature for $L = 20$. The results, as shown in Fig. 14, are consistent. This also indicates that the choice of $a_\tau$ depends more on the interaction strength $U$ rather than on the system size $L$. Therefore, for computing the phase diagram, it is safe to set $a_\tau t$ to be 0.05.

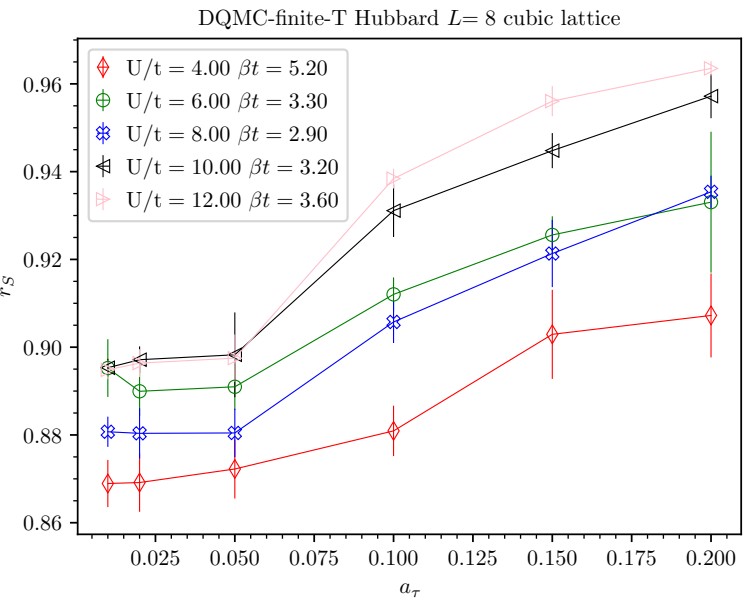

Figure 13: The relationship between $r_S$ and $a_\tau$ in DQMC-finite-T of the Hubbard model on a cubic lattice when $L = 8$.

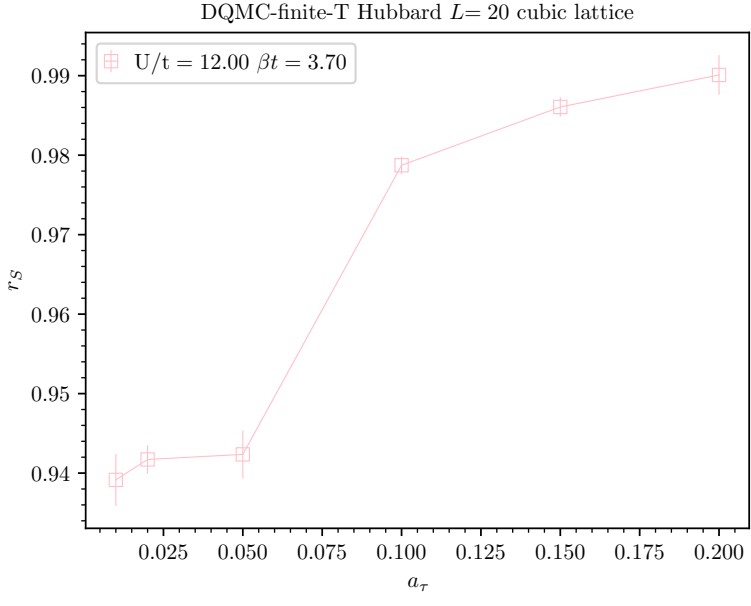

Figure 14: The relationship between $r_S$ and $a_\tau$ in DQMC-finite-T of the Hubbard model on a cubic lattice when $L = 20$ and $U/t = 12.00$.

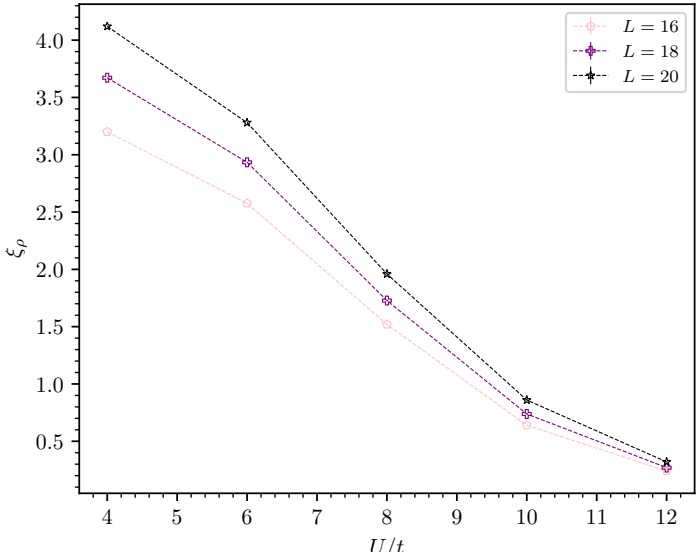

Figure 15: The relationship between the charge correlation length $\xi_\rho$ and $U/t$ in DQMC-finite-T of the Hubbard model on a cubic lattice near the critical temperature when $L = 16, 18$ and $20$.

## G   The O(3) universality class and finite-size effect

Our numeric results support that the finite temperature Néel transition is a continuous transition and belongs to O(3) universality class. This conclusion stems from our analysis at the finite temperature Néel transition point, where, although the fermion is gapless at the single-particle level, it exhibits a gap at the many-particle level, evidenced by the charge-charge correlation's exponential decay with distance. Furthermore, the many-body fermion charge gap strongly depends on the interaction strength - specifically, smaller $U$ values correspond to smaller gaps, which consequently induce larger finite-size effects.

This theoretical framework is robustly supported by our analysis of the charge correlation length. Our calculations demonstrate that the charge-charge correlation consistently decays exponentially with distance. To quantify this behavior, we calculate the charge correlation length for different $U$ and system sizes. The charge correlation length can be calculated as [80]:

$$\xi_\rho \equiv \frac{L}{2\pi}\sqrt{\frac{\rho(\mathbf{Q})}{\rho(\mathbf{Q}+d\mathbf{q})}-1}, \tag{G.1}$$

where $\rho(\mathbf{q})$ is the charge structure factor

$$\rho(\mathbf{q}) \equiv \frac{1}{N}\sum_{i,j}e^{i\mathbf{q}\cdot(\mathbf{r}_i-\mathbf{r}_j)}\left(\langle n_i n_j\rangle - \langle n_i\rangle\langle n_j\rangle\right), \tag{G.2}$$

with $d\mathbf{q} = (\frac{2\pi}{L}, 0, 0)$ and $\mathbf{Q} = (\pi, \pi, \pi)$. We systematically analyzed the charge correlation length for system sizes $L = 16$, $L = 18$, and $L = 20$ near the critical temperature for various values of $U/t$. The results are shown in Fig. 15. Our findings reveal that as the interaction strength decreases, the charge correlation length increases, demonstrating a smaller charge gap and subsequently stronger finite-size effects.

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
