# Peer review of "Boosting Determinant Quantum Monte Carlo with Submatrix Updates: Unveiling the Phase Diagram of the 3D Hubbard Model"

_SciPost Physics, doi:SciPost Phys. 18, 055 (2025)_

## Round 1 · Referee Report · Anonymous (Referee 1) · 2024-11-11

Report

The manuscript “Boosting Determinant Quantum Monte Carlo with Submatrix Updates: Unveiling the Phase Diagram of the 3D Hubbard Model” presents recent improvements to the Determinant Quantum Monte Carlo (DQMC) method. These improvements make it possible to simulate larger systems. In terms of physics results, the manuscript presents data for the half-filled 3D Hubbard model (when the sign problem is absent). The phase diagram of the half-filled Hubbard model has been extensively studied and the present paper does not shed new light on it. The main contribution is to simulate larger systems, enabling the authors to give a more accurate determination of the Néel temperature.

The core of the paper is fairly technical, aimed at specialists who want to know the details of these recent improvements. The article deals with three different update schemes: fast updates, delayed updates and submatrix updates. One of the main points of the manuscript is the implementation of submatrix updates

Although the article is very technical, providing such technical details is important for any researcher who wants to write their own DQMC code. Furthermore, understanding the physics of the Hubbard model is an important challenge in condensed-matter physics. It has become very clear in recent times that such an understanding goes hand in hand with technical developments / improvements in numerical techniques. That's why I am recommending publication. But before fully accepting the manuscript, I would like the following points to be addressed:

-The authors call their “innovative approach” a “generalized submatrix updating algorithm” because it can handle extended interactions and zero temperature. However, for the standard finite-temperature Hubbard model, which is the main model studied here, the introduction does not clearly explain, in my opinion, what has been done before and what is new. What's the difference with previous submatrix updating schemes for the finite-T half-filled Hubbard model (that couldn't go up to such large lattice sizes)? This is not clear from the current version of the manuscript. -On page 3, the authors write “As this additional overhead is implemented with Level 1 BLAS, its time cost can surpass the time cost for the update of Green’s function in practical calculation if one increase nd, as shown in Fig. 4.” But for the values of $n_d$ shown in Fig. 4, this doesn't seem to be the case. That's why I found this sentence somewhat confusing.
- It seems that the lattice sizes that were reached prior to this work were large enough to obtain a good estimate of the Néel temperature (using finite size scaling by crossing analysis). It is mentioned that previously one could go up to $N=1000$, or $L=10$. It's not very clear from the figures that going significantly beyond L=10 implies a significant improvement in the calculation of T_c. Could the authors indicate how much moving to larger lattice sites has decreased the error bar on $T_c$, which is what matters in the end. A related question: are there other quantities for which the effect of finite size is more important? -The paper focuses on a static property of the Hubbard model, in order to determine the Néel temperature. What about dynamical properties of the system? Is it expected that one could also go up to $L=20$?

Recommendation

Ask for minor revision

  • validity: -
  • significance: -
  • originality: -
  • clarity: -
  • formatting: -
  • grammar: -

Author:  Fanjie Sun  on 2024-12-25  [id 5067]

(in reply to Report 1 on 2024-11-11)
Category:
answer to question

Many thanks to you for your valuable feedback. We have addressed these comments and revised the manuscript accordingly. Please see the attached file for details.

Attachment:

reply1_7t0un3J.pdf

---

## Round 1 · Referee Report · Anonymous (Referee 2) · 2024-11-28

Strengths

  1. Detailed account of state of the art implementation of auxiliary field quantum Monte Carlo.

Weaknesses

  1. There are no obvious weaknesses

Report

The efficiency of BLAS routines depends on the ratio of floating-point operations to memory references. Level 1 BLAS corresponding to vector-vector operations are slower than Level 3 BLAS routines corresponding to matrix multiplication. In the realm of fermion Monte Carlo methods delayed updates as well as block updates have been developed in the past. The idea is to carry out less Level 1 BLAS at the expense of more level 3 BLAS operations. Overall, one carries out more flowing point approximations, but the code runs quicker! The article describes these known methods very well in the realm of the auxiliary field quantum Monte Carlo algorithm, both projective and finite temperature approaches. With these optimizations schemes the authors can reach lattices up to 8000 sites. This is however not the record since using similar algorithms but to the best of my knowledge different optimizations scheme the authors of Phys. Rev. B 99, 125145 (2019) were able to reach 10000 sites albeit at rather weak coupling.

As for the application, the authors look into the Hubbard model at half-filling on the cubic lattice, and determine the Néel temperature in terms as a function of the Hubbard U. I agree that the universality class is the 3D O(3) one. However, I think that the paper could benefit from a discussion of why this is so. The point is that at T_N the charge degrees of freedom are not gaped such as that one has to find an argument why they can be omitted in the critical theory. My naive understanding is that at T_N the charge excitations decay exponentially such that one can safely omit them in the critical theory. Given this conjecture, one can understand why corrections to scaling are more important in the weak coupling limit. This would stem for the fact that at weak coupling T_N is smaller, there is no pseudo gap, and the charge correlation length at T_N is longer at weak
than at intermediate coupling.

I think that a discussion along these lines, as well as some numerical data that computes the charge correlation length as a function of U/t could enhance the impact of the paper and would certainly provide a better understanding of why the authors expect the transition to belong to the 3D O(3) universality class.

Requested changes

1- Add reference Phys. Rev. B 99, 125145 (2019) 2- Discuss why it is so obvious that the O(3) universality class is the correct universality class. 3- Discuss why corrections to scaling seem to be more important at small values of U/t. 4- It would be very useful to compute the charge correlation length at T_N. Maybe that you have the data. If so, I think that it would be nice to include.

Recommendation

Ask for minor revision

  • validity: good
  • significance: good
  • originality: ok
  • clarity: high
  • formatting: perfect
  • grammar: perfect

Author:  Fanjie Sun  on 2024-12-25  [id 5066]

(in reply to Report 2 on 2024-11-28)
Category:
answer to question

Many thanks to you for your valuable feedback. We have addressed these comments and revised the manuscript accordingly. Please see the attached file for details.

Attachment:

reply1.pdf

---

## Round 2 · Referee Report · Anonymous (Referee 2) · 2024-12-27

Report

The authors have taken into account my comments. I think that the paper has progressed.

Requested changes

None.

Recommendation

Publish (easily meets expectations and criteria for this Journal; among top 50%)

---

## Round 2 · Referee Report · Anonymous (Referee 1) · 2025-1-7

Report

The authors have answered my questions and have taken into account my comments.

Requested changes

None.

Recommendation

Publish (easily meets expectations and criteria for this Journal; among top 50%)

---

## Round 2 · Author Response

Dear Editor,
Thank you very much for the editorial assessment and forwarding the report from the respected referees on November 28th.
Below are our point-to-point responses to the suggestions from the respected referees, and we have made corresponding changes in the revised manuscript. With these, we would humbly like to resubmit our manuscript to SciPost Physics.

Sincerely,
Fanjie Sun and Xiao Yan Xu

---

## Round 2 · List of Changes

According to the respected referees’ comments, we made the following revisions.
• We make our contribution in the abstract and introduction clearer.
• We have added reference Phys. Rev. B 99, 125145 (2019).
• We have added the appendix G to explain the universality class and finite-size effects of the model.
• Some typos are fixed.

---

## Editorial Decision

published